# Glucagon-Like Peptide-1 Receptor Agonist Attenuates Autophagy to Ameliorate Pulmonary Arterial Hypertension through Drp1/NOX- and Atg-5/Atg-7/Beclin-1/LC3β Pathways

**DOI:** 10.3390/ijms20143435

**Published:** 2019-07-12

**Authors:** Yi-Chia Wu, Wei-Ting Wang, Su-Shin Lee, Yur-Ren Kuo, Ya-Chin Wang, Shih-Jung Yen, Mei-Yueh Lee, Jwu-Lai Yeh

**Affiliations:** 1Division of Plastic Surgery, Department of Surgery, Kaohsiung Medical University Hospital, Kaohsiung 80708, Taiwan; 2Ph.D. Program in Translational Medicine, Kaohsiung Medical University and Academia Sinica, Taipei 11564, Taiwan; 3Research Center of Regenerative Medicine and Cell Therapy, Kaohsiung Medical University, Kaohsiung 80708, Taiwan; 4Translational Research Center, Department of Medical Research, Kaohsiung Medical University Hospital, Kaohsiung 80708, Taiwan; 5Center of Teaching and Research, Kaohsiung Municipal Ta-Tung Hospital, Kaohsiung 80145, Taiwan; 6Department of Surgery, School of Medicine, College of Medicine, Kaohsiung Medical University, Kaohsiung 80708, Taiwan; 7Department of Internal Medicine, Kaohsiung Municipal Ta-Tung Hospital, Kaohsiung 80145, Taiwan; 8Division of Endocrinology and Metabolism, Department of Internal Medicine, Kaohsiung Medical University Hospital, 807 Kaohsiung 80708, Taiwan; 9Department of Pharmacology, School of Medicine, College of Medicine, Kaohsiung Medical University, Kaohsiung 80708, Taiwan; 10Department of Medical Research, Kaohsiung Medical University Hospital, Kaohsiung 80708, Taiwan

**Keywords:** GLP-1 receptor agonists, liraglutide, pulmonary artery hypertension, autophagy, mitochondria

## Abstract

Mitochondrial dysfunction is associated with cardiovascular diseases and diabetes. Pulmonary arterial hypertension (PAH) is characterized by pulmonary vascular remodeling, and the abnormal proliferation, apoptosis and migration of pulmonary arterial smooth muscle cells (PASMCs). The glucagon-like peptide-1 (GLP-1) receptor agonist, liraglutide, has been shown to prevent pulmonary hypertension in monocrotaline-exposed rats. The aim of this study was to investigate the effect of liraglutide on autophagy, mitochondrial stress and apoptosis induced by platelet-derived growth factor BB (PDGF-BB). PASMCs were exposed to PDGF-BB, and changes in mitochondrial morphology, fusion-associated protein markers, and reactive oxygen species (ROS) production were examined. Autophagy was assessed according to the expressions of microtubule-associated protein light chain 3 (LC3)-II, LC3 puncta and Beclin-1. Western blot analysis was used to assess apoptosis, mitochondrial stress and autophagy markers. Liraglutide significantly inhibited PDGF-BB proliferation, migration and motility in PASMCs. PDGF-BB-induced ROS production was mitigated by liraglutide. Liraglutide increased the expression of α-smooth muscle actin (α-SMA) and decreased the expression of p-Yes-associated protein (p-YAP), inhibited autophagy-related protein (Atg)-5, Atg-7, Beclin-1 and the formation of LC3-β and mitochondrial fusion protein dynamin-related (Drp)1. Therefore, liraglutide can mitigate the proliferation of PASMCs via inhibiting cellular Drp1/nicotinamide adenine dinucleotide phosphate (NADPH) oxidases (NOX) pathways and Atg-5/Atg-7/Beclin-1/LC3β-dependent pathways of autophagy in PAH.

## 1. Introduction

Mitochondria consist of double membrane bound organelles, play a fundamental role in cell physiology, and undergo oxidative phosphorylation to produce ATP. Mitochondria contain nearly 80 different polypeptides encompassing five transmembrane protein complexes that direct electrons through the electron transport chain, and they are also involved in apoptosis and the generation of free radicals [1]. Therefore, mitochondrial function in cells is very important for many cellular physiological processes, including energy balance, cell differentiation and death, metabolic signaling, intracellular calcium ion balance, and lipid biosynthesis [2]. Recent studies have shown the importance of mitochondria in the cell cycle and cell growth [2,3]. Mitochondria have been implicated in many diseases, including mitochondrial disorders, age-related neurodegenerative diseases such as Parkinson’s disease and Alzheimer’s disease, pulmonary hypertension, arrhythmia, heart failure and cancer, as well as immune-related sepsis, intestinal-related autism, obesity, and diabetes [4,5]. In addition, damage to the mitochondria has been reported to lead to lipid degeneration and insulin resistance in the liver [6]. Although the mechanisms by which the mitochondria are involved in diseases are not fully understood, new therapeutic agents are urgently needed. Several studies have shown that loss of mitochondrial function can lead to a series of subsequent effects, including mitochondrial respiratory chain dysregulation, excessive free radical production, and excitotoxicity, which are important factors leading to disease [7,8]. Reactive oxygen and nitrogen species are signaling molecules in low levels, but in high levels, they damage organelles, particularly the mitochondria. Oxidative damage and the associated mitochondrial dysfunction may result in energy depletion, accumulation of cytotoxic mediators and cell death. Recent studies have found that one major sensor of redox signaling at this switch in cellular responses is autophagy [9].

Autophagy is the process by which dysfunctional cellular components are degraded inside the cell through the action of lysosomes. The process of autophagy involves the formation of double membrane vesicles that enwrap portions of the cytoplasm. The process consists of the following stages: (a) induction and phagophore formation; (b) autophagosome elongation and formation; and (c) fusion, degradation and recycling [10]. Autophagy is a dynamic process which is present in all cells at low levels under basal conditions, but stimuli such as nutrient starvation or hypoxia can lead to its upregulation. Autophagy is a tightly regulated pathway which at basal level has an important housekeeping role allowing cells to survive in response to multiple stress conditions [11,12]. Autophagy has been widely implicated in many pathophysiological processes such as cancer, metabolic and neurodegenerative disorders as well as cardiovascular and pulmonary diseases. It also has an important role in aging and exercise [13]. Autophagic activities are mediated by a complex molecular machinery including more than 30 Atg proteins and 50 lysosomal hydrolases. Autophagosomes form membrane structures, sequester damaged, oxidized or dysfunctional intracellular components and organelles, and direct them to the lysosomes for degradation. This autophagic process is the sole known mechanism for mitochondrial turnover. It has been speculated that dysfunction of autophagy may result in abnormal mitochondrial function and oxidative or nitrative stress [14].

Pulmonary arterial hypertension (PAH), the occlusive vascular disease, has several mitochondrial abnormalities, especially in the aerobic glycolytic and mitochondrial fragmentation. Occlusion of the blood vessels increases the afterload of the right ventricle, leading to right ventricular failure. Normally, mitochondria in the pulmonary artery smooth muscle cells act as oxygen receptors and patients with PAH have abnormal mitochondria. Inhibiting mitochondrial division or promoting mitochondrial fusion in the pulmonary arterial smooth muscle cells (PASMCs) of patients with PAH has been shown to lead to cell proliferation, cell cycle arrest, and apoptosis [15].

Glucagon-like peptide-1 (GLP-1) is a derivative of the proglucagon gene transgenic product, which is mainly synthesized by L cells of the ileal mucosa. When food enters the gut, GLP-1 is released and acts on pancreatic beta cells, stimulating insulin secretion. GLP-1 can bind to the GLP-1 receptor (GLP-1R), which is mainly expressed in the pancreas, heart, blood vessels, gastrointestinal tract, kidneys, lungs, breasts, and central nervous system. The physiological functions of GLP-1 are mainly the stimulation of insulin secretion, slowing down islet cell apoptosis, and increasing islet cell mass. GLP-1 has also been shown to have an extra-pancreatic effect that inhibits gastric emptying and reduces food intake [16,17]. GLP-1 receptor agonists have been studied in recent years on cardiovascular disease and endothelial dysfunction [18], and its treatment protects endothelial cells from oxidative stress-induced autophagy and endothelial dysfunction [19]. Liraglutide suppressing endothelin-1 (ET-1) and enhancing endothelium NO synthase (eNOS)/soluble guanylate cyclase (sGC)/protein kinase G (PKG) pathways were also studied in animal models of monocrotaline-induced PAH [20].

Mitochondrial dynamics include transitions between elongated interconnected mitochondrial networks and fragmentation and re-arrangement caused by mitochondrial fusion and fragmentation. Mitochondrial division is regulated by the mitochondrial fission 1 (Fis1) and dynamin-related (Drp1) proteins. Fis1 is present in the outer membrane of the mitochondria and absorbs cytoplasmic Drp1 during cell division, thereby triggering contraction of the mitochondrial membrane. In contrast, fusion is controlled by the large GTPase fusion molecules mitofusins 1/2 (MFN 1/2) and first optic atrophic protein (Opa1), which promote fusion of the outer and inner membranes, respectively [21]. The effects of liraglutide on mitochondrial structure, reactive oxygen species (ROS) production, membrane potential surface, the phosphorylation of Drp1 protein during mitochondrial fusion, MFN2, and the respiratory chain are unclear. In this study, we would like to investigate the mechanism of liraglutide on the dynamic regulation of mitochondria and its effect on the morphological transformation of vascular smooth muscle cells in PAH, which could prevent proliferation of PASMCs by inhibiting cellular PASMC migration and phenotype switching to improve mitochondrial function and autophagy function, thereby ameliorating PAH.

## 2. Results

### 2.1. Liraglutide Inhibited Platelet-Derived Growth Factor BB (PDGF-BB)-Induced PASMC Proliferation

After 24, 48, and 72 h incubation in the absence or presence of liraglutide (1, 5, and 10 nM), cells were harvested, and viability was examined using the 3-(4,5-dimethylthiazol-2-yl)-2,5-diphenyltetrazolium bromide (MTT) assay. Liraglutide decreased PASMC viability in a concentration-dependent manner at 24, 48, and 72 h. This effect was reversed by the GLP-1R antagonist exendin (9–39) (500 nM) (Figure 1).

### 2.2. Liraglutide Inhibited PDGF-BB-Induced PASMC Migration and Motility

To elucidate the anti-migratory effects of liraglutide, we performed two types of migration assays. We first examined whether liraglutide pretreatment (1, 5, and 10 nM, 1 h) affected PDGF-BB (20 ng/mL)-induced PASMC migration using a Boyden chamber assay. As shown in Figure 2A, PDGF-BB induced migration of the PASMCs into the upper chamber from the lower chamber at 24 h, and this PDGF-BB-induced cell migration was significantly inhibited by liraglutide in a concentration-dependent manner. This effect was reversed by exendin (9–39) (500 nM). We then assessed the inhibitory effect of liraglutide on cell migration in the artificial scratch wound of PDGF-BB-stimulated PASMCs. As shown in Figure 2B, liraglutide significantly reduced the number of cells in the scratch wound area of PDGF-BB-induced PASMCs in a concentration-dependent manner, and this effect was similarly reversed by exendin (9–39) (500 nM).

### 2.3. Liraglutide Attenuated PDGF-BB-Induced Mitochondrial ROS Generation

To clarify the relationship of PDGF-BB and liraglutide in ROS regulation in PASMCs, we performed the specific ROS probe dichlorodihydrofluorescein diacetate (DCFH-DA) for ROS detection. Fluoroscopy showed that PDGF-BB (20 ng/mL) induced ROS expression in PASMCs. However, this PDGF-induced ROS production was reduced due to liraglutide treatment in a dose-dependent manner (Figure 3).

### 2.4. Liraglutide Attenuated PDGF-BB-Induced Cell Apoptosis

We next examined whether PDGF-BB (20 ng/mL) was able to result in cell apoptosis in PASMCs. As shown in Figure 4, cells treated with PDGF-BB showed apparent green fluorescence by using tetraethylbenzimidazolylcarbocyanine iodide (JC-1) staining indicating an imbalance of mitochondrial membrane potential, which usually occurs in apoptotic cells. However, when the cells were pretreated with liraglutide, the emission of green fluorescence triggered by PDGF-BB was restored. Cells treated with liraglutide exhibited strong red fluorescence (healthy and non-apoptotic cells) in a dose dependent manner instead of green signals (Figure 4).

### 2.5. Liraglutide Attenuated PDGF-BB-Induced NADPH Oxidase 1 (NOX1) Expression and Mitochondrial Fission in the PASMCs

In addition to mitochondria, nicotinamide adenine dinucleotide phosphate (NADPH) oxidases (NOX) are regarded as major sources of ROS in the vasculature. Among the NOX family, NOX1 is particularly implicated in proliferative vascular diseases such as atherosclerosis and hypertension. Accordingly, we investigated the effect of monocrotaline (MCT)-induced PAH PASMCs on PDGF-BB-induced NOX1 expression. We found that liraglutide downregulated PDGF-BB-induced NOX1 expression, and that this effect was blocked by the GLP-1R antagonist exendin (9–39). Previous studies have shown that migration of Drp1 from the cytosol to the mitochondrial surface is an initial step in mitochondrial fission [22]. Therefore, we evaluated whether mitochondrial fusion triggered by liraglutide was associated with changes in the subcellular distribution of Drp1. Western blotting showed that the expression of activated Drp1 (phosphorylated at Ser 616) was upregulated in MCT-induced PAH PASMCs. The phosphorylation of Drp1 was decreased after 24 h of treatment with liraglutide 10 nM in the PAH PASMCs, and these effects were blocked by the GLP-1R antagonist exendin (9–39) (Figure 5).

### 2.6. Liraglutide Inhibited PDGF-BB-Induced Phenotype Switching in the PASMCs

Expressions of the contractile and synthetic phenotypes were measured in cells treated with liraglutide (1, 5, 10 nM), and PDGF-BB-induced phenotype switching in the PASMCs was assessed. As shown in Figure 6A,B, PDGF treatment resulted in an increase in p-Yes-associated protein (p-YAP) at 12 and 24 h and a decrease in α-smooth muscle actin (α-SMA) at 24 and 48 h. Liraglutide further downregulated the expression of p-YAP but upregulated the expression of SM 22α in PDGF-treated PASMCs (Figure 6C,D).

### 2.7. Liraglutide Attenuated PDGF-BB-Induced Autophagy in the PASMCs

We then assessed the effects of PDGF-BB on autophagy in the PASMCs by examining autophagy-related protein (Atg)-5, Atg-7, Beclin-1, and LC3-β formation. As shown in Figure 7A, PDGF-BB stimulation caused a time-dependent increase in Atg-7 and LC3-β formation at 6, 12 h, 24 h, and 48 h, and Atg-5 and Beclin-1 formation at 6 h, 12 h and 24 h, suggesting that PDGF promoted autophagy. The effects of PDGF-BB stimulation on Atg-5, Atg-7, Beclin-1, and LC3-β formation were attenuated by liraglutide 10 nM. These effects were reversed by exendin (9–39) and 3-methyladenine (3-MA), which is an inhibitor of phosphatidylinositol 3-kinase (PI3K), a key regulator of autophagy in controlling the activation of mammalian target of rapamycin (mTOR) (Figure 7B).

## 3. Discussion

In this study, the GLP-1 receptor agonist liraglutide inhibited PDGF-BB-induced PASMC proliferation, migration, and motility. Liraglutide also attenuated PDGF-BB-induced mitochondrial ROS generation and mitochondrial membrane potential electrical charge imbalance, PDGF-BB-induced NOX1 expression, and mitochondrial fission Drp1 in PASMCs. In addition, it attenuated autophagy by Atg-5, Atg-7, Beclin-1, and LC3-β formation, and increased p-YAP and decreased α-SMA to inhibit PDGF-BB-induced phenotype switching in the PASMCs (Figure 8).

Endogenous GLP-1 has a half-life of only a few minutes and is then rapidly degraded by dipeptidyl peptidase 4 (DPP-4) [16]. Therefore, more stable exendin-based GLP-1, GLP-1 human analogues, and DPP-4 inhibitors have been developed as therapeutic drugs for the treatment of type 2 diabetes. Several studies have shown that GLP-1R agonists and their analogues have potent cardiovascular protective effects and functions, prompting many scientific and clinical studies on their antihypertensive effects. Exendin-based GLP-1 exenatide has been shown to reduce weight and glycosylated hemoglobin (HbA1c), systolic blood pressure, triglycerides, and high-sensitivity C-reactive protein (CRP) (hsCRP) in obese patients with type 2 diabetes. In addition, its effect on systolic blood pressure was confirmed in a pooled analysis of 2171 patients [23]. The long-acting GLP-1R agonist, liraglutide, has been shown to have significant effects with regard to weight loss and reducing systolic blood pressure in Asian subjects [24]. Antihypertensive effects have also been observed with DPP-4 inhibitors [25]. The effects of these antidiabetic agents are of great interest, especially those with extra pancreatic effects on the cardiovascular system, given the additional advantages of cardiovascular protection through the different but related complex mechanisms of these medications [26].

Autophagy and apoptosis occur in a number of disease conditions, including atherosclerosis, hypertension, myocardial ischemia, and certain tumors [27,28]. During conditions of nutrient deficiency, cells circulate through autophagy to reuse basic biomolecules, remove damaged organelles and harmful proteins, and eliminate intracellular pathogens. Therefore, the process of autophagy plays a key role in maintaining intracellular homeostasis and contributes to cell survival during periods of cell stress [29,30]. Recent studies have shown that autophagy dysfunction is associated with lung diseases, and especially chronic obstructive pulmonary diseases such as cigarette smoke-induced emphysema. In addition, several diseases associated with autophagy are thought to be associated with this autophagic imbalance, and this can affect the development and duration of these diseases [31,32].

Many studies have indicated that pathological hyperplasia of pulmonary vascular smooth muscle cells and mitogen-activated protein kinase (MAPK) and AKT pathways are related [33]. The PDGF signaling system consists of PDGF-A, PDGF-B, PDGF-C, and PDGF-D ligands and PDGF receptor (PDGFR)-α and PDGFR-β receptors [34]. PDGF-BB has been shown to induce pulmonary vascular smooth muscle cell hyperplasia, atherosclerosis, pulmonary fibrosis, pulmonary hypertension, and pulmonary embolism caused by chronic embolism [35]. In addition, increased levels of PDGF have been reported in the blood and lung tissues of patients with pulmonary hypertension, further confirming that PDGF plays a key role in pulmonary vascular remodeling and increased pulmonary artery pressure [36]. In our previous study, we found that liraglutide could inhibit the proliferation and migration of vascular smooth muscle cells induced by PDGF-BB, and reduce cell division and proliferation, through inhibiting the activation mechanism of PAH via eNOS/NO/cGMP/PKG after the induction of MCT [20]. In this study, liraglutide attenuates autophagy to ameliorate PAH through Drp1/NOX- and Atg-5/Atg-7/Beclin-1/LC3β pathways.

Some studies have discussed the biological effects of autophagy in regulating smooth muscle cells. It has been hypothesized that when vascular cells are under pressure, autophagy plays an important role in vascular modulation and regulating the transformation of smooth muscle cell morphology [36]. Heterogeneous nuclear ribonucleoprotein E1 as a new regulator of vascular endothelial cells apoptosis and autophagy through mediating homeobox-containing 1 (HMBOX1) expression, opened the door to a novel therapeutic drug for related vascular diseases [37]. The vascular remodeling and increased pulmonary vascular resistance in PAH can eventually lead to right ventricular failure and death [38]. The abnormal proliferation of PASMCs can cause vascular remodeling and occlusion, an important feature of pulmonary hypertension [39]. Therefore, it is important to clarify the specific molecular mechanisms and signal transmission pathways of pulmonary artery vascular smooth muscle hyperplasia.

In a study of idiopathic PAH, the expression of LC3B-II in mature autophagosomes was found to be higher than in controls. Although the mechanism by which autophagy affects pulmonary hypertension is not clear, it still appears to play an important role in revascularization [38]. In a study of LC3B-II gene knockout mice, pulmonary hypertension was shown to be caused by chronic hypoxia, suggesting that autophagy is increased in hypoxic environments in pulmonary artery endothelial cells as a protective mechanism [40]. LC3-II is a marker of Atg-5/Atg-7-dependent autophagy. Studies have demonstrated that autophagy depends on Atg-5/Atg-7, that it is associated with microtubule-associated protein LC3 lipidation and truncation, and that it may arise from the membrane of the endoplasmic reticulum and other membrane organelles [41]. Beclin-1 has been shown to play a vital role in Atg-5- and Atg-7-dependent and -independent autophagy. In addition, it has been shown to modulate Vps-34 protein, a lipid kinase, and induce formation of Beclin-1-Vps34-Vps15 core complexes through interactions with various cofactors, consequently promoting autophagy [42]. In addition, the cleavage of Beclin-1 mediated by caspase has been shown to induce crosstalk between autophagy and apoptosis, and Beclin-1 dysfunction has been reported in disorders such as cancer and neurodegenerative diseases [43]. Chloroquine, the traditional treatment for malaria and rheumatoid arthritis, has been shown to inhibit autophagy and pulmonary hypertension induced by MCT, and to increase autophagy in pulmonary vascular smooth muscle cells [44].

In recent years, studies on GLP-1 and autophagy have shown that liraglutide can protect insulin-secreting pancreatic beta cells (INS-1 cells) from apoptosis induced by high glucose and prevent islet β cell apoptosis by autophagy [43]. The autophagy protein ATG5 plays an essential role in the formation of the autophagosome via promoting the lipidation of microtubule-associated protein LC3, which is involved in the expansion and completion of the autophagosome [45]. In an animal study, the expression of Atg-5 and conversion of LC3-II to LC3-I were significantly enhanced by liraglutide, indicating that liraglutide enhanced autophagy in a high-fat diet and streptomycin-induced type 2 diabetic mice [46].

The present study provides a potential explanation of the vascular protective effects of the GLP-1 receptor agonist liraglutide, including stimulating mitochondrial fusion, increasing mitochondrial activity, and decreasing PDGF-BB-induced PASMC dedifferentiation. These results indicate that liraglutide enhances functional coupling between endoplasmic reticulum and mitochondria pathways. Our data also suggest that liraglutide inhibits vascular remodeling through a mitochondrial dynamic-dependent mechanism. Taken together, these findings demonstrate that an increased GLP-1 level may have beneficial and protective effects on the progression of vascular diseases. Therefore, liraglutide may be a potential therapeutic target for patients with PAH.

## 4. Materials and Methods

### 4.1. Animals

All animal care and experimental protocols in this study were approved by the Animal Care and Use Committee of Kaohsiung Medical University (Permit No. 105188, 27 January 2017). Eight-week-old male Wistar rats (200–250 g) were provided by the National Laboratory Animal Breeding and Research Center (Taipei, Taiwan), and were housed under a constant temperature and controlled illumination. Food and water were available ad libitum.

### 4.2. Preparation of PASMCs

The development of PAH was examined in the rats after a single injection of monocrotaline (MCT; 60 mg/kg, s.c.) for 21 days. MCT was dissolved in 0.5 N HCl and adjusted to pH 7.4 with 0.5 N NaOH solution [10,11,12]. Normal Wistar rats and those with PAH were sacrificed with an overdose of sodium pentobarbital (60 mg/kg), and the skin was sterilized with 75% alcohol. The chests of the mice were opened, and the hearts and lungs were removed and rinsed several times in phosphate-buffered saline (PBS). The pulmonary arteries were dissected under sterile conditions, and the outer spheres were peeled. The microtubules were then cut, and the endothelium was gently shaved two to three times to remove endothelial cells. The tunica media was cut into 1-mm^3^ pieces in Dulbecco’s Modified Eagle’s Medium (DMEM). The PASMCs were then cultured in DMEM containing 10% fetal bovine serum (FBS) (5% CO_2_ at 37 °C). The culture medium was changed every three days, and the cells were subcultured until confluence. Primary cultures of two to four passages were used in the experiments. The PASMCs were examined by immunofluorescence staining of α-actin to confirm their purity. Over 95% of the cell preparations were found to be composed of smooth muscle cells.

### 4.3. Cell Proliferation Assay

The 3-(4,5-dimethylthiazol-2-yl)-2,5-diphenyltetrazolium bromide (MTT) assay was used to assess proliferation of the PASMCs. In brief, after serum starvation for 48 h, proliferation of the PASMCs was induced by PDGF-BB (20 ng/mL) in DMEM supplemented with 1% FBS in 96-well plates at a density of 2 × 10^4^ cells per well, with or without 1–10 nM liraglutide (Novo Nordisk, Novo Alle, Bagsvaerd, Denmark) pre-treatment (1 h). The GLP-1 receptor antagonist 500 nM exendin (9–39) (Sigma–Aldrich, St. Louis, MO, USA) was added to the cells for 30 min prior to treatment with liraglutide. After induction for 24 h, 48 h, and 72 h, MTT (0.5 mg/mL) was added to the medium for 4 h. The culture medium was then removed, and the cells were dissolved in isopropanol and shaken for 10 min. The amount of MTT formazan was quantified at 540 and 630 nm using an enzyme-linked immunosorbent assay (ELISA) reader (DYNEX Technologies, Denkendorf, Germany).

### 4.4. Determination of Cell Migration

Two migration assays were also performed. First, the PDGF-BB-mediated PASMC migration assay was performed using a Boyden chamber. Briefly, PASMCs (2 × 10^4^) were loaded into the upper compartment, while PDGF-BB (20 ng/mL) was dissolved in serum-free DMEM with various concentrations of liraglutide (1–10 nM, pre-treatment for 1 h) in the lower compartment. After 24 h, the non-migrated cells on the surface of the upper membrane were removed, and the cells on the lower surface were fixed in methanol and subjected to Giemsa staining. The number of cells in six high-power fields (HPF; 200×) was counted, with the mean number being used to assess migration activity.

PASMCs that were grown to confluence in 6-well cell culture plates were used for the wounding assay. FBS was removed from the medium, and serum-free medium was used. A clean wound area was produced with a pipette tip 24 h after serum depletion. Photographs of the wounds were then obtained 24 h after wounding in the serum-free medium (control) or in the presence of 20 ng/mL PDGF-BB. Liraglutide was added to the culture medium 1 h before adding PDGF-BB, and its effect was analyzed. The distance from the leading edge of the cells that had migrated to the edge of the wound was recorded for analysis.

### 4.5. Autophagy Assay

PASMCs were placed in a hypoxic chamber for 30 min in serum-free medium with or without liraglutide and Ex9-39, and then reperfused (with medium containing 10% serum) with or without liraglutide for 2 h. A competitive antagonist of GLP1 receptor agonist (GLP-1RA), exendin (9–39) (500 nM), was added 30 min before liraglutide treatment. Control cells were treated with 2.5 mM 3-methyladenine (3-MA) (Sigma–Aldrich, St. Louis, MO, USA), a known inhibitor of autophagy, for 30 min.

### 4.6. Western Blot Analysis

PASMCs were lysed with cell lysis buffer containing 20 mmol/L Tris (pH 7.5), 150 mmol/L NaCl, 1% Triton X-100, sodium pyrophosphate, β-glycerophosphate, EDTA, Na_3_VO_4_, leupeptin, and 1 mmol/L phenylmethanesulfonyl fluoride (PMSF). The protein concentration was then determined using a bicinchoninic acid assay (BCA). Equal amounts of protein were separated on SDS-polyacrylamide gels and probed with specific antibodies against NOX-1, contractile and synthetic phenotype proteins p-YAP and SM 22α, microtubule-associated protein 1A/1B-light chain 3 (LC3)-II, Beclin-1, Atg-5 and Atg-7, and mitochondrial fission protein DRP-1 (Cell Signaling Technology). The membranes were probed with the antibodies at 4 °C overnight and blocked with a solution containing 5% nonfat milk at room temperature for 1 h. The membranes were then washed with Tris-buffered saline Tween-20 (TBST) buffer for 5 min and incubated with horseradish peroxidase conjugated antibodies at room temperature for 1 h. After washing again with TBST buffer for 5 min, immunolabeled bands were detected by electrochemiluminescence. β-actin was used as an internal control.

### 4.7. Determination of Intracellular ROS

To measure PDGF-BB-induced ROS generation, PASMCs (1 × 105 cells/well) were treated with PDGF-BB in hypoxic chambers in the presence or absence of liraglutide (1, 10, and 100 nM) for 6 h [47]. The cells were then incubated with 10 μM of dichlorodihydrofluorescein diacetate (DCFH-DA) for 30 min at 37 °C. Nonfluorescent DCFH-DA is able to pass through the cell membrane and be hydrolyzed by esterase to form non-penetrating DCFH and is further oxidized by intracellular ROS to fluorescent DCF. Therefore, the accumulation of DCF can be used to indicate the level of intracellular ROS. Furthermore, since camptothecin is well recognized to exhibit cytotoxic effects by increasing ROS generation, we used camptothecin as a positive control in this study.

### 4.8. Determination of Mitochondrial Membrane Potential

To determine mitochondrial membrane potential, tetraethylbenzimidazolylcarbocyanine iodide (JC-1), a cationic dye was prepared and used to determine the mitochondrial membrane potential. Since JC-1 is positively charged, it is attracted by the negative electric field of the mitochondrial membrane and aggregate to form a dimer and emit orange fluorescence when it enters the cell. On the other hand, once the mitochondrial membrane potential disappears, the JC-1 dye keeps the monomer state with green fluorescence. In this study, PASMCs were cultured on chamber slides for the experimental treatments. Afterwards, the cells were incubated with 5 μg/mL JC-1 for 30 min at 37 °C in a 5% CO_2_ incubator. The level of JC-1 fluorescence was then examined and recorded using a fluorescent microscope for later photographic analysis.

### 4.9. Statistical Analysis

The results are expressed as mean ± standard error of the mean (SEM). Statistical differences were estimated using the Student–Newman–Keuls method and one-way analysis of variance (ANOVA) followed by Dunnett’s test for comparisons of group means. A *p*-value of 0.05 was considered to be significant.

## Figures and Tables

**Figure 1 ijms-20-03435-f001:**
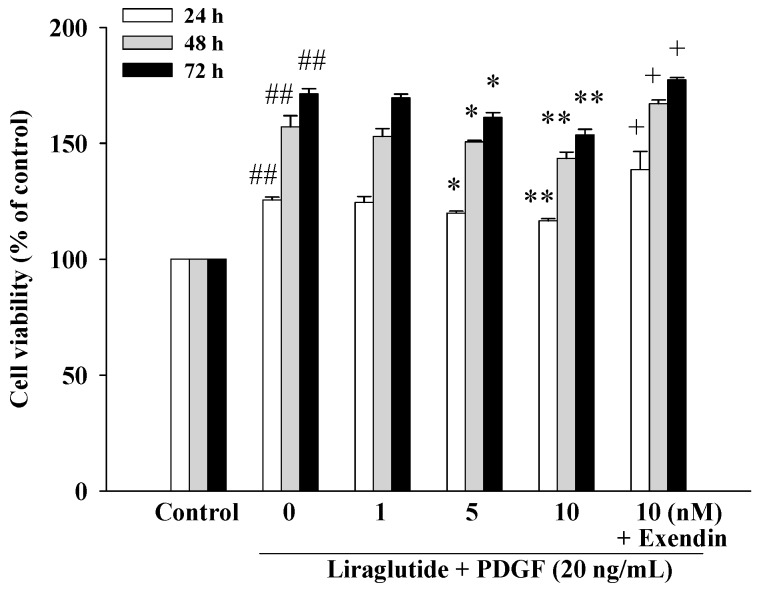
Effects of liraglutide on 20 ng/mL platelet-derived growth factor BB (PDGF-BB)-induced rat pulmonary arterial smooth muscle cell (PASMC) proliferation. After 24, 48, and 72 h incubation in the absence or presence of liraglutide (1–10 nM), cells were harvested, and viability was examined using the 3-(4,5-dimethylthiazol-2-yl)-2,5-diphenyltetrazolium bromide (MTT) test. Values represent the mean ± SEM of *n* = 6. Control: PASMCs were placed in medium with 1% fetal bovine serum (FBS). ^##^
*p* < 0.01 versus control group; * *p* < 0.05, ** *p* < 0.01 versus cells exposed to PDGF-BB alone. ^+^
*p* < 0.05 versus liraglutide (10 nM) + PDGF (20 ng/mL) group. Exendin, exendin (9–39).

**Figure 2 ijms-20-03435-f002:**
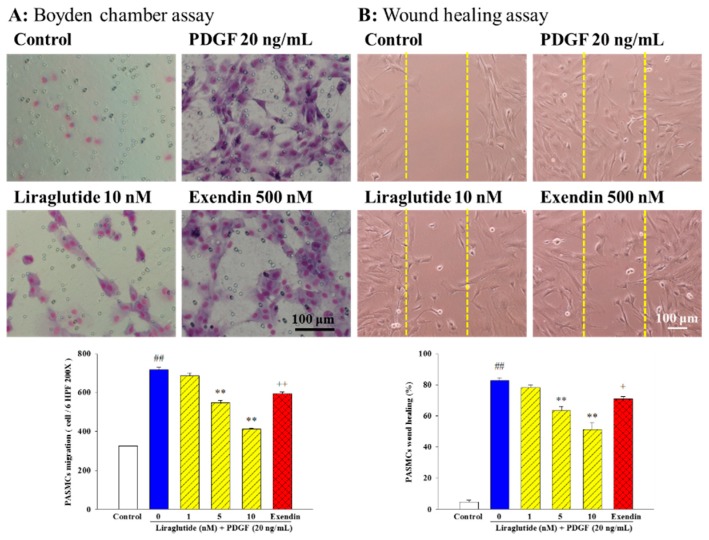
Effects of liraglutide on cell migration in Boyden chamber and wound healing assays. (**A**) PDGF-BB induced migration of rat PASMCs into the upper chamber from the lower chamber at 24 h. Liraglutide inhibited PDGF-BB-induced migration in a dose dependent manner. The bar graph shows migration activities assayed four times in three independent experiments for the number of cells observed in six high-power fields. (**B**) Wound closure was photographed (top) and measured under microscopy (high-power fields (HPF) ×100) (bottom) in a wound-healing assay. Scale bar = 100 μm. Values represent the mean ± SEM of *n* = 4. Control: PASMCs were placed in medium with 1% FBS. ^##^
*p* < 0.01 versus control group; ** *p* < 0.01 versus cells exposed to PDGF-BB alone; ^+^
*p* < 0.05, ^++^
*p* < 0.01 versus the liraglutide (10 nM) + PDGF (20 ng/mL) group. Exendin, exendin (9–39).

**Figure 3 ijms-20-03435-f003:**
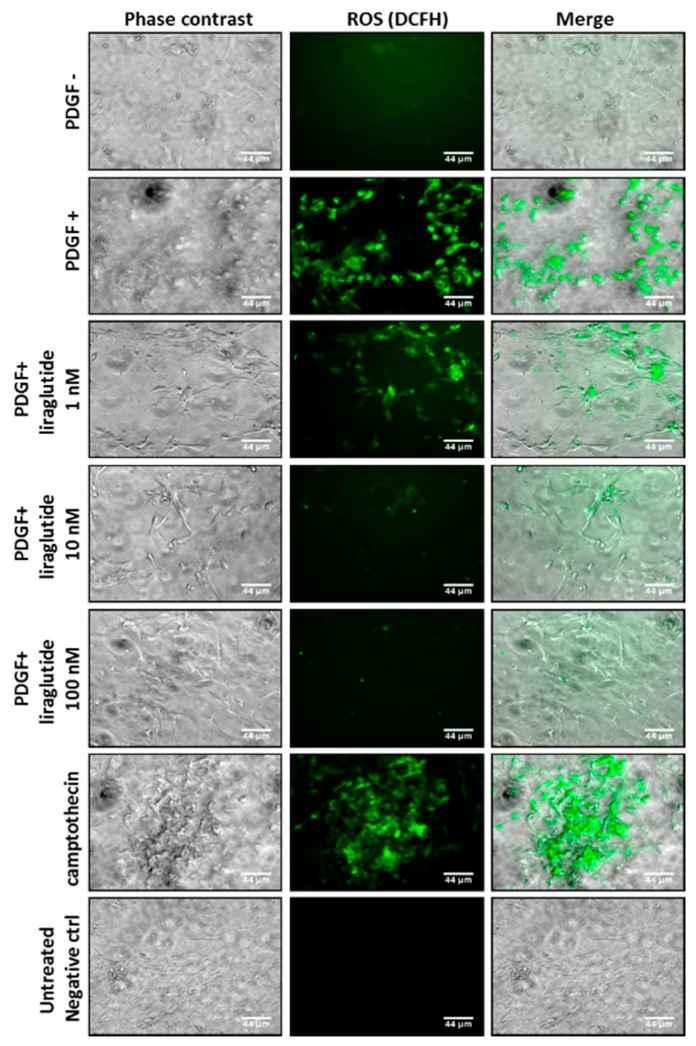
Liraglutide inhibited PDGF-BB-induced mitochondrial reactive oxygen species (ROS) production. By detecting the fluorescence of dichlorodihydrofluorescein (DCF), the level of intracellular ROS was detected. Camptothecin served as a positive control. PASMCs alone and those treated with different doses of liraglutide for 1 h were observed. Magnification: ×400. Scale bar = 44 μm.

**Figure 4 ijms-20-03435-f004:**
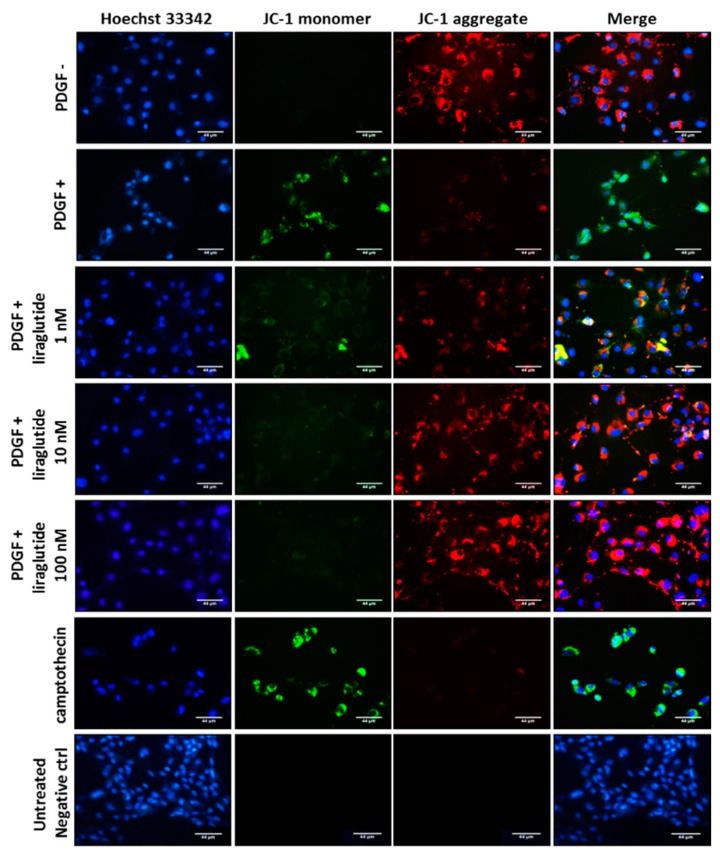
Liraglutide attenuated PDGF-BB-induced mitochondrial membrane potential electrical charge imbalance. Tetraethylbenzimidazolylcarbocyanine iodide (JC-1) is a cationic dye which can be used to measure mitochondrial membrane potential. The JC-1 stain can be used to observe apoptosis, emitting red fluorescence in healthy cells and green light when cell apoptosis occurs. By detecting the fluorescence of JC-1, the mitochondrial membrane potential electrical charge imbalance was detected. PASMCs alone and those treated with different doses of liraglutide for 1 h were observed. Magnification: ×400. Scale bar = 44 μm.

**Figure 5 ijms-20-03435-f005:**
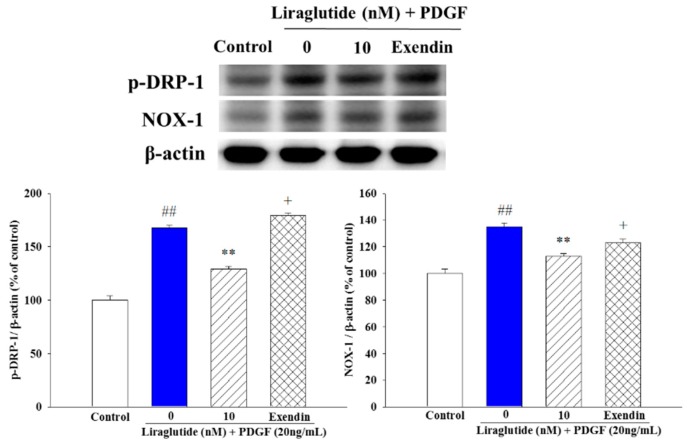
Effects of liraglutide on the PDGF-BB-induced activation of NADPH oxidase 1 (NOX1) and phosphorylation of dynamin-related (Drp1) in rat PASMCs. Liraglutide prevented PDGF-BB-induced NOX-1 activation and Drp1 activation in monocrotaline (MCT)-induced pulmonary arterial hypertension (PAH) PASMCs. Densitometric results represent the mean ± SEM of *n* = 4. ^##^
*p* < 0.01 versus control group; ** *p* < 0.01 versus cells exposed to PDGF-BB alone; ^+^
*p* < 0.05 versus the liraglutide (10 nM) + PDGF (20 ng/mL) group. Exendin, exendin (9–39).

**Figure 6 ijms-20-03435-f006:**
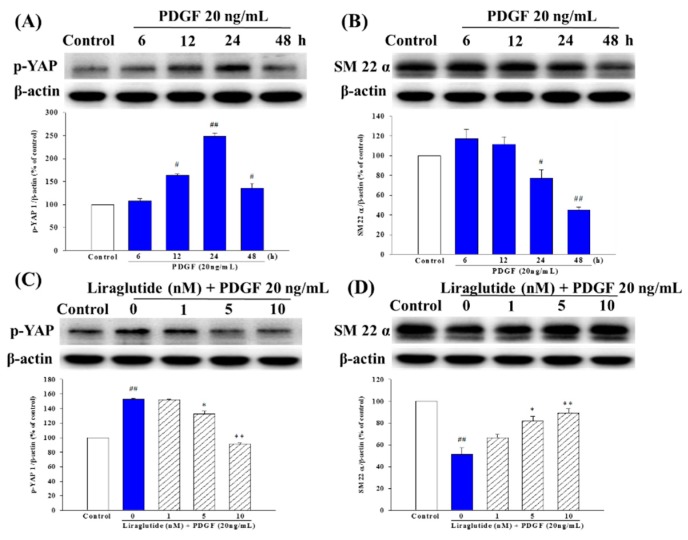
PDGF-BB induced a contractile-to-synthetic phenotype switch in PASMCs. Western blot analysis of contractile and synthetic proteins after PDGF-BB stimulation: PASMCs were serum-starved for 24 h and then stimulated with PDGF-BB from 6 to 48 h. (**A**,**C**) p-YAP was associated with the synthetic phenotype; (**B**,**D**) α-smooth muscle actin (α-SMA) is a marker of the contractile phenotype. PASMCs were treated with different doses of liraglutide for 1 h, followed by co-treatment with PDGF-BB (20 ng/mL) for 24 h. (**C**,**D**) Liraglutide prevented PDGF-BB-induced phenotype switching. Densitometric results represent the mean ± SEM of *n* = 4. ^#^
*p* < 0.05, ^##^
*p* < 0.01 versus control group; * *p* < 0.05, ** *p* < 0.01 versus cells exposed to PDGF-BB alone.

**Figure 7 ijms-20-03435-f007:**
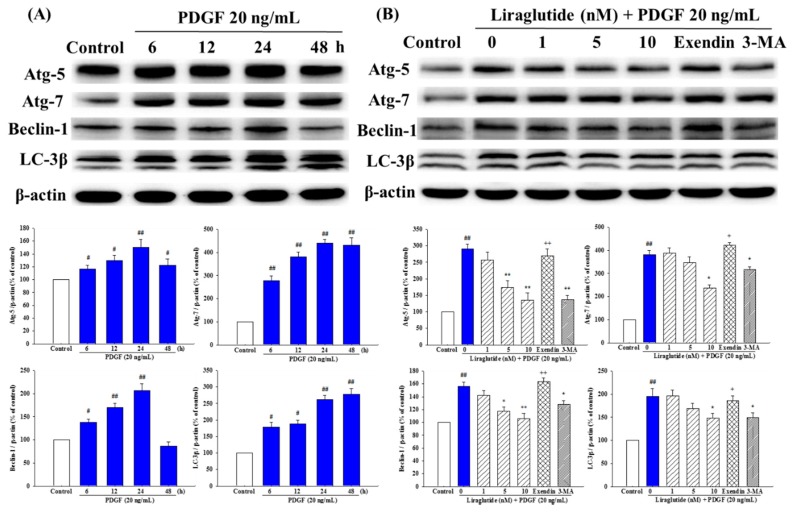
Effects of liraglutide on PDGF-BB-induced expressions of autophagy-related protein (Atg)-5, Atg-7, beclin-1, and LC3-II in rat PASMCs. (**A**) PDGF-BB induced autophagy in PASMCs. Western blot analysis of autophagy marker proteins after PDGF-BB stimulation. β-actin was used as an internal standard protein. (**B**) Liraglutide attenuated the expressions of PDGF-BB-induced autophagy markers. PDGF-BB was tested alone or in combination with liraglutide for 24 h. Densitometric results represent the mean ± SEM of *n* = 4. ^#^
*p* < 0.05, ^##^
*p* < 0.01 versus control group; * *p* < 0.05, ** *p* < 0.01 versus cells exposed to PDGF-BB alone; ^+^
*p* < 0.05, ^++^
*p* < 0.01 versus the liraglutide (10 nM) + PDGF-BB (20 ng/mL) group. Exendin, exendin (9–39).

**Figure 8 ijms-20-03435-f008:**
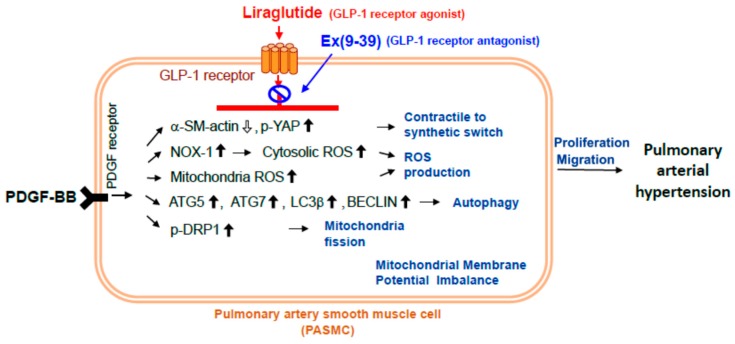
Proposed mechanism by which liraglutide ameliorates pulmonary arterial hypertension through Drp1/NOX and Atg-5/Atg-7/Beclin-1/LC3β-dependent autophagy pathways. Liraglutide inhibited PDGF-BB-induced PASMC proliferation, migration, and motility. Liraglutide also attenuated PDGF-BB-induced mitochondrial ROS generation and mitochondrial membrane potential electrical charge imbalance. An increase in p-YAP and decrease in α-SMA inhibited PDGF-BB-induced phenotype switching in PASMCs.

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
