# Peer review of "Glucagon-Like Peptide-1 Receptor Agonist Attenuates Autophagy to Ameliorate Pulmonary Arterial Hypertension through Drp1/NOX- and Atg-5/Atg-7/Beclin-1/LC3β Pathways"

_ijms, 2019, doi:10.3390/ijms20143435_

Round 1

Reviewer 1 Report

Wu et al analyze in their manuscript the effect of liraglutide, a glucagon-like peptide-1 receptor agonist, on autophagy in the context of pulmonary arterial hypertension. 

In general, the experiments themselves are sound. However, not each claim by the authors is represented in an experiment and the quality of the figures is poor.

1) There are several formatting mistakes in the text (e.g. underlined words in lines 64-65) and many different styles in the literature list.   

2) The quality of all figures is poor. This is caused by a low resolution. Please use figures with a higher resolution.

3) Please add a scale bar to the microscopy pictures.

4) explanation for Atg: different definitions in lines 49 and 90

5) The authors mention a "previous study" of their group without a citation (line 285).Please add a citation. Why do the results of this study differ from the study in Ref. 19?

6) The authors claim (starting line 324) that their present study provids explanations of the vascular protective effects of liraglutide. They mention an effect on mitochondrial fusion. Where is this shown in the paper? They mention increasing mitochondrialactivity. To which experiment and to which activity do they refer?

Author Response

Wu et al analyze in their manuscript the effect of liraglutide, a glucagon-like peptide-1 receptor agonist, on autophagy in the context of pulmonary arterial hypertension. 

In general, the experiments themselves are sound. However, not each claim by the authors is represented in an experiment and the quality of the figures is poor.

1)      There are several formatting mistakes in the text (e.g. underlined words in lines 64-65) and many different styles in the literature list.   

Ans: All the formatting mistakes were now corrected, especially the reference part and the underlined words in lines 64-65.

2)      The quality of all figures is poor. This is caused by a low resolution. Please use figures with a higher resolution.

Ans. We now provided a higher resolution of all figures after using high resolution software to do our best as we can.

3) Please add a scale bar to the microscopy pictures.

Ans. Scale bar of the microscopy pictures in Figures 2, 3, and 4, and in figure legends were added as your good suggestion.

4) explanation for Atg: different definitions in lines 49 and 90

 Ans. Since the autophagy (Atg) related proteins was already defined in line 49, to make the definition uniform, we just left the Atg in abbreviation in line 90.

5) The authors mention a "previous study" of their group without a citation (line 285). Please add a citation. Why do the results of this study differ from the study in Ref. 19?

Ans. The “previous study” description in DrP-1 was added with a reference (ref. 22) in line 196-197. Our study had shown GLP-1 receptor agonist attenuates autophagy to ameliorate one of the cardiovascular disease, the pulmonary arterial hypertension through Drp1/NOX- and Atg-5/Atg-7/Beclin-1/LC3β pathways, which is somehow different from the other study in the way on its treatment in protecting endothelial cells from oxidative stress-induced autophagy and endothelial dysfunction (ref. 19). But both studies have a common endpoint in protective effect on cardiovascular disease.

GLP-1 receptor agonists have been studied in recent years on cardiovascular disease and endothelial dysfunction [18], and its treatment protects endothelial cells from oxidative stress-induced autophagy and endothelial dysfunction [19]. Liraglutide suppressing ET-1 and enhancing eNOS/sGC/PKG pathways was also studied in animal model of monocrotaline-induced PAH [20].

Another “previous study” mentioned in line 293-296, we were referring to the same “our previous study” in ref. 20, and now we rephrase it. The previous study is different from our present study in the pathway involved in ameliorating PAH by liraglutide, one is suppressing ET-1 and enhancing eNOS/sGC/PKG pathways was also studied in animal model of monocrotaline-induced PAH ,and another one is attenuates autophagy to ameliorates PAH through Drp1/NOX- and Atg-5/Atg-7/Beclin-1/LC3β pathways. This statement was also added in line 293-297.

6) The authors claim (starting line 324) that their present study provids explanations of the vascular protective effects of liraglutide. They mention an effect on mitochondrial fusion. Where is this shown in the paper? They mention increasing mitochondrial activity. To which experiment and to which activity do they refer?

Ans. Mitochondrial division is regulated by the mitochondrial fission 1 (Fis1) and dynamin-related (Drp1) proteins, and we used Drp-1 protein to represent the effect of mitochondrial fission and fussion. This result was shown in Figure 5. The mitochondrial activity was described by Figure 4. As shown in Figure 4, cells treated with PDGF-BB showed apparent green fluorescence by using JC-1 staining indicating an imbalance of mitochondrial membrane potential, which usually occurs in apoptotic cells. However, when the cells were pretreated with liraglutide, the emission of green fluorescence triggered by PDGF-BB was restored. Cells treated with liraglutide exhibited strong red fluorescence (healthy and non-apoptotic cells) as a dose dependent manner instead of green signals.

Reviewer 2 Report

This manuscript describes study of the effects of liraglutide on the rat model for Pulmonary Arterial Hypertension aiming to provide a potential explanation of its vascular protective effects.

The results are very interesting and to some degree novel. Liraglutide has been shown to stimulate autophagy in previous studies related to its cardiovascular effects. There seems to be no introductory comment or discussion on this.

The title of the article is "Glucagon-Like Peptide-1 Receptor Agonist Enhances Autophagy to Ameliorates Pulmonary Arterial Hypertension through Drp1/NOX- and Atg-5/Atg 7/Beclin-1/LC3β Pathways"

Yet the abstract (which is correct) states that "inhibiting cellular Drp1/NOX pathways and Atg-5/Atg-7/Beclin-1/LC3β-dependent pathways of autophagy"

This seems to be a contradiction and needs to be corrected.

The figures are of low quality. There needs to used a different software program to export the images for the paper.

Author Response

This manuscript describes study of the effects of liraglutide on the rat model for Pulmonary Arterial Hypertension aiming to provide a potential explanation of its vascular protective effects.

The results are very interesting and to some degree novel. Liraglutide has been shown to stimulate autophagy in previous studies related to its cardiovascular effects. There seems to be no introductory comment or discussion on this.

The title of the article is "Glucagon-Like Peptide-1 Receptor Agonist Enhances Autophagy to Ameliorates Pulmonary Arterial Hypertension through Drp1/NOX- and Atg-5/Atg 7/Beclin-1/LC3β Pathways" 

Yet the abstract (which is correct) states that "inhibiting cellular Drp1/NOX pathways and Atg-5/Atg-7/Beclin-1/LC3β-dependent pathways of autophagy"

This seems to be a contradiction and needs to be corrected.

Ans.  Thank you for your correction. We already change the word “enhances” in the title to “attenuates”.  

The figures are of low quality. There needs to used a different software program to export the images for the paper.

Ans. We now provided a higher resolution figures after we did our best with our present available software program.

Round 2

Reviewer 1 Report

The authors have answered all my questions in the revised manuscript.